# Examining the Shift in the Decomposition Channel Structure of the Soil Decomposer Food Web: A Methods Comparison

**DOI:** 10.3390/microorganisms11102589

**Published:** 2023-10-19

**Authors:** Wen Xing, Ning Hu, Zhongfang Li, Meng Yuan, Meiling Luo, Shuo Han, Evgenia Blagodatskaya, Shunbao Lu, Yilai Lou

**Affiliations:** 1Institute of Environment and Sustainable Development in Agriculture, Chinese Academy of Agricultural Sciences, Beijing 100081, China; xingwen@caas.cn (W.X.); xiaomueng@163.com (M.Y.); luomeiling217@163.com (M.L.);; 2School of Food and Biological Engineering, Hezhou University, Hezhou 542899, China; 2009huning@163.com (N.H.); lizhongfang08@126.com (Z.L.); 3Department of Soil Ecology, Helmholtz Centre for Environmental Research—UFZ, 6108 Halle (Saale), Germany; janeblag@mail.ru; 4College of Life Sciences, Jiangxi Normal University, Nanchang 330022, China; luxunbao8012@126.com

**Keywords:** decomposition channel, fungi to bacteria ratio, fungivore to bacterivore ratio, glucosamine to muramic acid ratio, soil food web

## Abstract

Selecting the appropriate indicators and measuring time point numbers is important for accurately examining the shift in soil gross decomposition channel structure. Through a selected case study on a natural forest vs. rainfed arable system over a two-month-long experiment, the utility of three commonly employed indicators (fungi to bacteria ratio (F:B), fungivore to bacterivore ratio (FF:BF), and glucosamine to muramic acid ratio (GlcN:MurN)) were compared to reflect the shift in soil gross decomposition channel structure. The requirement of measuring the time point numbers for the three indicators was also assessed, and we suggest a potential methodology. Our results revealed that the GlcN:MurN ratio was more reliable for assessing the shifts in gross decomposition channel structure for long-term land use changes, while it was less sensitive to short-term drought compared with the other two indicators. The F:B ratio was more applicable than the FF:BF ratio for reflecting both long- and short-term changes. Furthermore, the reliability of the GlcN:MurN ratio was the least dependent on measuring time point numbers. We suggest the use of multiple indicators and the adoption of multiple measuring time points for the overall methodology.

## 1. Introduction

The soil decomposer food web is comprised of substrate resources (e.g., detritus and root exudates), microbial decomposers, microbivorous faunas, and the high trophic levels of predators [1,2,3,4]. Two energy channels of the soil food web can explain how changes in decomposition processes of detritus impact ecosystem processes. First, the bacterial-based channel includes bacteria as the primary decomposer of substrate resources, which mainly transfers the flow of energy and material through bacterial-feeding fauna and their predators at high trophic levels [5]. In general, the bacterial channel primarily occurs in nitrogen-rich soils that contain easily decomposable substrates, and promotes the nutrient mineralization rate and biological activities [5,6]. Second, the fungal-based channel includes fungi as the primary decomposer, which plays a key role in the exchange of matter and energy between substrate resources and the high trophic levels of fungal-feeding fauna and their predators [5]. The fungal channel derives most of its energy from recalcitrant organic matter (e.g., high carbon-to-nitrogen ratio) or acidic soil, and is linked to the highly conservative cycling of nutrients and increased soil carbon and nitrogen retention [7,8,9]. The importance of the fungal-based channel in the soil food web often increases following land abandonment or vegetation restoration [10,11,12,13], while the bacterial-based channel becomes increasingly important following intense disturbances, nutrient enrichment, and intensive farming [9,14,15]. Therefore, assessing the relative importance of fungal- vs. bacterial-based energy channels in soil decomposer food webs may be an explicit management objective for agricultural production practices or environmental protection programs. Although many previous studies have used the abundance or biomass of microbial and nematode functional groups and their metabolites to assess the structures of soil decomposer food webs, challenges remain for selecting the appropriate indicators to accurately examine the shifts in soil gross decomposition channels.

Three types of ecological indicators, based on (i) consumers, (ii) predators, and (iii) metabolites, can be used to reflect shifts in soil gross decomposition channel ratios. Firstly, for primary consumers, the fungi to bacteria (F:B) ratio is estimated by quantitative real-time polymerase chain reaction (qPCR) or phosphor lipid fatty acid (PLFA), which is the most widely applied indicator [16,17,18]. For instance, PLFA analysis provides specific markers for bacteria and fungi, which allows for the indirect separation of fundamentally different basal resources and the trophic chains that they support, that is, the bacterial and fungal energy channels in soils [19]. For the high trophic level, since the feeding habits of nematodes can be clearly inferred based on their oral structures, the fungal- to bacterial-feeding nematode abundance (FF:BF) ratio has been used as an indicator of decomposition pathways [20,21,22,23]. Thirdly, metabolite indicators such as amino sugars, which are reliable microbial residue biomarkers, can also uncover shifts in soil decomposition pathways [24]. Among the identified amino sugars, glucosamine acid (GlcN) is predominantly derived from fungal cell walls, while muramic acid (MurN) originates exclusively from bacterial cell walls [25,26]. As amino sugars are relatively biochemically stable in soil, the glucosamine to muramic acid (GlcN:MurN) ratio might provide a time-integrated measure of microbial responses, which has the potential to offer insights into the relative contributions of fungi vs. bacteria to detritus decomposition processes [24,27]. Although these indicators have been used extensively (either individually or in combination in previous studies), their reliability and the contexts of their applications toward the interpretation of soil food web characteristic have not been fully assessed.

There are at least two uncertainties regarding the reliability and application of these indicators. Firstly, there is no unified strategy for the selection of suitable indicators to infer the shifts in decomposition channel ratios of the soil food web [19]. However, the importance of biotic and abiotic indicators for predicting changes in soil decomposition channel ratios over the long vs. short term could be disparate in diverse ecosystems. For example, biotic indicators (F:B and FF:BF ratios) have the potential to uncover transient changes in soil decomposition channel ratios, whereas the GlcN:MurN ratio could provide more data regarding the overall states of decomposition processes after ecosystems have experienced long-term disturbances [19,24,28]. Secondly, the number of measuring time points for both biotic and abiotic indicators are often limited and random across different studies. Little is known about how many time points should be adopted to accurately predict changes in soil decomposition channel ratios over the long vs. short term [28]. To trace long-term shifts in the decomposition channel ratios of the soil food web, a few measuring time points for the GlcN:MurN ratio may be sufficient, as amino sugars are biochemically stable and typically not easily decomposed in nature [24]. In contrast, it is essential to employ multiple time points for quantifying biotic indicators, since soil organisms are very sensitive to specific climate events. For example, compared with fungi and fungal-feeding nematodes, bacteria and bacterial-feeding nematodes are more susceptible to fluctuations in soil temperature and moisture under transient drought events, which can dramatically shift bacterial-dominant energy channels to fungal-dominant energy channels [9,29]. For these reasons, the indicators utilized to disentangle the relative contributions of fungal and bacterial channels demand thorough evaluation. The indiscriminate use of these different methodological indicators may lead to the misinterpretation of anthropogenic impacts on soil biological processes, and further, incorrect management decisions.

For the present study, the reliability of both biotic (F:B and FF:BF ratios) and abiotic (GlcN:MurN ratio) indicators were assessed to predict the shifts in soil decomposition channel ratios between a natural forest and rainfed crop system with a long-term land use history (>60 years of cultivation). Further, the constant dynamics of these indicators during a two-month-long experiment were monitored to evaluate the shifts in energy channel ratios between the two ecosystems in response to changes in environmental conditions. We aimed to address the following questions: (1) How do different biotic and abiotic indictors predict shifts in the soil decomposition channel ratios between a natural forest and a rainfed crop system over the long vs. short term? (2) What are the requirements for measuring time point numbers of each indicator? We tested two hypotheses: (H1) The GlcN:MurN ratio provides a time-integrated measure of the fungal to bacterial channel ratio to predict changes between the two ecosystems over the long-term, while the F:B and FF:BF ratios offer more specific data on the decomposition channel ratio to predict their changes over the short term. (H2) A few measuring time points for the GlcN:MurN ratio are adequate for the detection of long-term shifts in the soil decomposition channel ratio between a natural forest and rainfed crop system, since amino sugars are negligibly decomposed in nature. In contrast, multiple time points are required for measuring soil decomposition channel ratios over short-term study periods, as many soil organisms are very sensitive to fluctuations in environmental conditions.

## 2. Materials and Methods

### 2.1. Case Study Site and Experiment Design

This case study was conducted in 2017 in Jianping County (42°03 N, 119°29 E), of Jiangxi Province, in Northeast China. This region belongs to a subtropical continental monsoon climate, with a mean annual precipitation of 450 mm and temperature of 6.5 °C. Two adjacent sites (each 600 m^2^) were selected for this study (a natural forest system (domain pine tree species) and rainfed crop (maize) system (over 60 years duration)), which represented contrasting ecosystems. This was because forest soil decomposition has been extensively shown to be more fungi-dominant relative to arable soil. Our selected study period spanned June and July, due to the strong rainfall fluctuations that occur during these months. The soil of the two sites was classified as Haplic-Ustic Cambisol (FAO classification). At each site, three plots (each 100 m^2^) were randomly selected as replicates. To focus on the detritus food web and exclude rooting effects, in situ 120 PVC collars (5 cm diameter × 10 cm depth) were randomly installed in each plot two months prior to sampling, with their openings above the soil surface to facilitate rainwater runoff.

### 2.2. Soil Sampling and Measurement

Soil samples within the PVC collars were collected in the morning (10:00 a.m.) every 2–4 d through June and July (for a total of 20 sampling time points). The soil moisture (*v*/*v*) and temperature (°C) were monitored by a sensor. The soil of five randomly selected collar points was manually composited and homogenized to prepare the samples (yielding a total of 60 samples, 3 replicates × 20 time points) from each site. All samples were immediately put on ice, transferred to the laboratory, divided into three parts, and stored at 4 °C, –80 °C, and under air-dried conditions, respectively.

Gene copies of the soil bacterial and fungal communities were determined, as were the bacterivore and fungivore abundances and the glucosamine and muramic acid contents. The soil genome DNA was extracted from 0.5 g of frozen soil using the FastDNA^TM^ SPIN Kit for Soil (MP Biomedicals, Santa Ana, CA, USA). Quantitative real-time polymerase chain reaction (qPCR) was performed to estimate the gene copies of the bacterial 16S and fungal ITS rRNA, using the bacterial primer set 27f/519R [30] and fungal primer set ITS1/4 [31]. All detailed qPCR procedures were carried out following the instructions described previously [32]. Three replicates without DNA samples were used as negative controls for primers. Soil nematodes were extracted from 100 g fresh soil via the cotton wool filter method [33]. After fixing with 4% formaldehyde at 65 °C, the nematodes were counted using a microscope. A minimum of 100 individuals were identified into trophic groups according to their feeding habits. The nematode abundance was expressed as individuals per 100 g of dry soil. The air-dried subsamples were sieved at <0.25 mm and pretreated for amino sugar analysis [34]. Briefly, the soil was hydrolyzed with 6 M HCl for 8 h, after which the solution was filtered, adjusted to pH 6.6–6.8, centrifuged, and freeze-dried. Methanol was added to remove the amino sugars from the residues and again freeze-dried. The purified amino sugars were converted to aldononitrile derivatives and extracted with dichloromethane from the aqueous solution. After evaporating the dichloromethane, the amino sugar derivatives were redissolved in the mixed hexane and ethyl acetate solvent for quantification. The amino sugar derivatives were detected using Agilent 6890A Gas Chromatography (Agilent Tech. Co., Palo Alto, CA, USA). Glucosamine and muramic acid were considered as fungal and bacterial markers, respectively. Gene copies of the fungi to bacteria (F:B) ratio, abundance of bacterivore to fungivore (FF:BF) ratio, and content of glucosamine to muramic acid (GlcN:MurN) ratio were calculated as three indicators for the decomposition channel ratio.

The subsamples collected on 16 June and 16 July were used for the following soil properties assay. The soil organic carbon (SOC) and total nitrogen (TN) contents were quantified using an element analyzer (Vario EL, Elementar, Hanau, Germany) after excluding inorganic C by an acid treatment. Next, the soil C:N was calculated as the ratio of SOC to TN, after which the KMnO_4_-oxidized C was determined [35]. Briefly, the soil (sieved at 0.25 mm) containing 15 mg C was placed into 50 mL centrifuge tubes to which 25 mL of 333 mM KMnO_4_ were added. The centrifuge tubes were shaken for 6 h and centrifuged for 5 min at 2000 rpm. The absorbance of the supernatant and standards were read at 565 nm. Changes in the KMnO_4_ concentration were used to estimate the amount of oxidized C, under the assumption that 1 mM KMnO_4_ was consumed during the oxidation of 0.75 mM or 9 g of C. The decomposability of the SOC was assessed by the proportion of KMnO_4_-oxidized C to SOC. The soil pH was measured with the supernatants at a 1:2.5 (soil mass: water volume) ratio using a pH meter. The soil aggregate composition was determined by the wet-sieving method. The proportion of >0.25 mm aggregates (R_0.25_) was quantified after correcting the weight by the sand content. The soil porosity was determined by the core method. Briefly, an undisturbed core soil (100 cm^3^) was oven-dried and weighed, after which the bulk density (BD, g/cm^3^) was calculated. The soil porosity (%) was then estimated using the equation: 1 − BD/2.65.

### 2.3. Statistical Analysis

All statistical analyses were performed in R version 4.1.1 (R Development Core Team 2021). It was assumed that the forest soil gross decomposition channeled toward fungal dominance, relative to the crop soil [9,36]. Thus, the effective indication meant that the average value of the indicator was significantly higher for the forest soil than the crop soil at a specific time point. The differences in F:B, FF:BF, and GlcN:MurN ratios between the two studied systems at each measuring time point, respectively, were evaluated using the T test. If an indicator showed effectiveness over most of the time points across a total of 20, it was considered as reliable for indicating the shifts in decomposition channel structure under long-term land use changes. If an indicator was ineffective during drought periods, this meant that it was sensitive to short-term drought fluctuations.

The reliability of each indicator in revealing long-term land use changes was then quantified using a random selection method. Specifically, (N = 1, 2, …, 20) time points were randomly selected from a total time span across 20. The number of all combinations of N randomly selected time points was C_20_^N^, and all corresponding combinations were collected using the *combn* function in the *utils* package in R. For any indicator, we defined an applicable utilization for a combination as that where the indicator effectively indicated a shift in the gross channel ratio at >50% of all selected time points (as statistically detected by the T-test mentioned above). Subsequently, all numbers of combinations with an applicable utilization (CAU_num_) were collected under 1, 2, 3, …, 20 randomly selected time points, respectively. The reliability was then calculated as follows: reliability = CAU_num_/C_20_^N^ × 100%. The minimum number that fully (reliability = 100%) indicated a shift in the decomposition channel ratio was considered the appropriate requirement number of measured time points.

The differences between the soil physicochemical variables of the two studied systems were also evaluated using the *t* test.

## 3. Results

### 3.1. Soil Moisture and Temperature

In our case study, the soil moisture was consistently higher for the natural forest in contrast to the rainfed crop system, and exhibited a temporal fluctuation with the lowest values during two drought periods (i.e., 25–28 June and 11–13 July) (Figure 1a). No significant differences in the soil temperature between forest and crop systems were detected (Figure 1b).

### 3.2. Responses of the Three Indicators to Long-Term Land Use Changes

In general, all the three indicators (particularly F:B and GlcN:MurN ratios) were sensitive in response to long-term land use changes (Figure 2). At most measuring time points, the values of both F:B and GlcN:MurN ratios for the forest soil were significantly higher than those for the rainfed crop soil. This supported a general view that the forest soil gross decomposition was relatively dominated by fungal-based channels in contrast to the rainfed crop soil.

A significantly higher F:B ratio in the forest soil was found at 14 measuring time points compared with the crop soil, while there was no significant difference in the F:B ratio at 6 time points (Figure 2a). A significantly higher FF:BF ratio in the forest soil was observed at fewer time points (11) (Figure 2b). The GlcN:MurN ratio responses exhibited the greatest temporal stability, which was higher in the forest soil than the crop soil at most (up to 19) measuring time points (Figure 2c).

### 3.3. Quantified Reliability under a Series of Randomly Measuring Time Point Number

Under any number of randomly selected measuring time points, the GlcN:MurN ratio had the highest reliability, followed by the F:B ratio, and the FF:BF ratio had the worst reliability in reflecting the shifts in the gross decomposition channel ratio in response to long-term land use changes (Figure 3). The reliability of the GlcN:MurN ratio was the least dependent on the randomly selected measuring time point numbers between the three indicators. Overall, the reliability of both the F:B and GlcN:MurN ratios increased with higher random time point numbers. The required minimum numbers of randomly selected measuring time points were 13 and 3 for the F:B and GlcN:MurN ratios, respectively, with 100% reliability in reflecting the shifts in gross decomposition channel ratio in response to land use changes (Figure 3).

### 3.4. Selected Key Soil Abiotic Properties

Higher SOC contents, TN contents, SOC:TN ratios, labile C contents (KMnO_4_-oxidized C), R_0.25_, and porosity values, and a lower SOC decomposability value were observed in the forest soil in contrast to the rainfed crop soil (Figure 4). There were no differences in the soil pH between the two studied land uses.

## 4. Discussion

### 4.1. GlcN:MurN Ratio Was Most Reliable Indicator of Shifts in Gross Decomposition Channel Ratio in Response to Long-Term Land Use Changes

We observed that the average values of F:B, FF:BF, and GluN:GlcN ratios in natural forest soil were higher than for rainfed crop soil during the two-month-long experimental period. This suggested that the three indicators confirmed the dominance of the fungal decomposition channel in the natural forest relative to the rainfed crop system. Substrate resources, soil physical properties, and soil moisture largely affected shifts in the decomposition channel ratio [12,19,22,37]. More recalcitrant substrate qualities (i.e., lower C:N ratio) in natural forest soil favored the growth of fungi, while a higher proportion of aggregates and porosity provided better microhabitats for the expansion of fungal hyphae, thus leading to a higher F:B ratio in the forest soil [38,39]. However, as we observed in this study, the forest soil moisture was generally higher than the crop soil, which favored bacterial-based decomposition channel relative to fungi and resulted in a lower F:B ratio [9]. Consequently, the gross difference in decomposition channel between the contrasting systems under study was primarily determined by substrate quality, soil aggregation, and soil porosity (i.e., the general contributions of these factors were greater than soil moisture). It should be noted that the effect of soil moisture was dependent on the moisture range [40,41]. We predicted that the impacts of soil moisture during drought might be greater, and may even offset the effects of other factors, which might alter the decomposition channel ratio differences between the two systems. Thus, estimations of the general shifts in decomposition channel ratio based on instantaneous measurements during drought might lead to errors.

No significant differences in the F:B ratios between the natural forest soil and rainfed crop soil were found at six time points (especially during two drought periods), which was consistent with our first hypothesis. This calls into question the common use of randomly limited measuring time points of F:B ratio indicators to assess general shifts in the decomposition channel ratio of long-term land use changes. The F:B ratio serves as an instantaneous indicator at a specific time point, which can be strongly affected by environmental conditions [18,19,42]. During drier periods, the opposite impacts of soil moisture were strong enough to offset the effects of other factors; thus, the F:B ratio remained constant within the forest and crop systems. Therefore, our study indicated that the F:B ratio response to long-term treatments was time-dependent.

As for the nematode indicator, the insignificant responses of the FF:BF ratio were observed at more time points than those of the F:B ratio (i.e., 11 vs. 6), which contributed to the lower reliability of the FF:BF than the F:B ratio. This also called for caution when using only the FF:BF ratio indicator to examine the general shifts in the decomposition channel ratio of long-term land use changes. Notably, the anticipated responses of the FF:BF ratio were delayed relative to the F:B ratio responses to the altering systems during both drought periods. This might be explained by nematodes having a longer turnover time than microorganisms. More time is required to change nematode abundance after shifts in their soil microorganism prey [28]. Modified soil moisture and porosity might also directly influence nematodes and in turn reduce their responses to microorganisms [43,44]. We concluded that the FF:BF ratio was less reliable than the F:B ratio for evaluating shifts in the gross decomposition channel ratio of long-term treatments.

Consistent with our first hypothesis, the GlcN:MurN ratio was the most reliable indicator for shifts in gross decomposition channel ratio in response to long-term land use changes. This was indicated by both the number of effective time points (up to 19 of a total of 20) and the highest reliability quantified at any randomly measured time point. Significant differences in the GlcN:MurN ratios between the natural forest and rainfed crop soils were consistent, even during drought periods. Thus, residue indicators could reflect microbial processes across a longer timeline, during which the decomposition channel ratio was still dominantly controlled by substrate quality and soil porosity (not offset by moisture effects). Theoretically, microbial residue indicators are relatively stable and less sensitive to changes in environmental factors than F:B and FF:BF ratios. Therefore, they may reflect variations in historical decomposition channel status over long time scales [26,45]. Further, a recent study suggested that microbial residues may contribute to about half of the accumulated soil organic matter [46]. These high background amounts may have significantly counteracted the minor changes in amino sugar quantities observed during our study period, making changes in the GlcN:MurN ratio difficult to detect. Taken together, our study confirmed that the microbial residue GlcN:MurN ratio indicator was more reliable than those indicators at both the consumer and predator levels for evaluating shifts in gross decomposition channel ratio in response to long-term treatments (summarized in Figure 5).

### 4.2. The F:B Ratio Was More Reliable than FF:BF and GlcN:MurN Ratios in Response to Short-Term Drought

As discussed above, both the F:B and FF:BF ratios were more sensitive than the GlcN:MurN ratio to short-term drought in reflecting shifts in the decomposition channel ratio between natural forest and rainfed crop soils. Furthermore, especially for the crop soil alone, the values of both F:B and FF:BF ratios were obviously higher during drought than non-drought periods, while the GlcN:MurN ratio remained stable. This was because drought can inhibit microbial growth and activities, while fungi is thought to be more resistant to drought than bacteria. Therefore, soil gross decomposition channels can be assumed to shift toward fungal dominance in response to drought [9,47]. However, the F:B ratio was more reliable than the FF:BF ratio due to the response delay of FF:BF during drought. Thus, the above results suggested that the F:B ratio indicator was the most appliable for the short-term scales of the study, and to avoid using the FF:BF ratio, or especially the GlcN:MurN ratio alone (summarized in Figure 5).

### 4.3. The GlcN:MurN Ratio Was Least Dependent on the Number of Measured Time Points

The minimum number of time points required to fully indicate shifts in decomposition channel ratio for each indicator may be adopted as a recommendation for the suitable number of measuring time points for similar studies that are focused on long-term land use changes. The minimum number of time points for the F:B ratio was up to 13. In fact, few field studies have adopted this many measuring time points [7,17,18]. For the FF:BF ratio, the minimum number of time points required to fully indicate shifts in the decomposition channel ratio was not found with 1–20 time points. Consistent with our second hypothesis, for the GlcN:MurN ratio, at least three time points were sufficient to reflect shifts in the decomposition channel ratio. Overall, based on the minimum number of time points, the GlcN:MurN ratio proved to be the most applicable indicator for long-term study treatments (summarized in Figure 5).

## 5. Conclusions

Based on a natural forest and rainfed crop system (with >60 years of cultivation), our results demonstrated that the GlcN:MurN ratio was reliable for assessing shifts in soil decomposition channel structure under long-term land use changes, using relatively few measuring time points. However, the F:B ratio was more reliable than the FF:BF ratio for reflecting brief changes in the soil decomposer food web structure following short-term environmental changes.

Due to the limitations of the applied indicators for this study, the reliability of other indicators estimated by the PLFA, growth rate, carbon metabolism, and specific nematode functional guilds require further assessment. We suggest the use of diverse indicators and adoption of multiple measuring time points for an overall methodology. First, future research should assess the mechanisms through which changes in decomposition channel structure relate to the question of interest. Second, the most relevant strategy for determining fungal vs. bacterial channel dominance should be selected. Specifically, to interpret changes in decomposition pathways over long-term scales, the amino sugar index is likely a more applicable indicator with few required measurement time points. However, to interpret changes in decomposition pathways over short-term scales, biotic indicators such as the fungi to bacteria ratio using multiple measurement time points are likely a prudent choice to reflect changes in the soil decomposer food web.

## Figures and Tables

**Figure 1 microorganisms-11-02589-f001:**
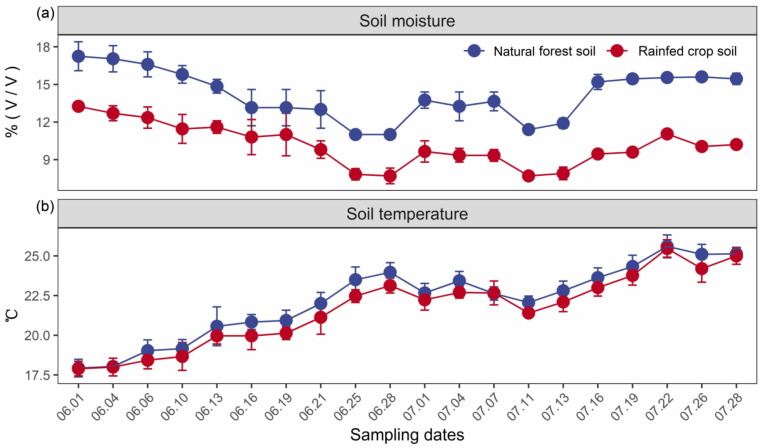
Temporal changes in soil moisture (*v*/*v*) (**a**) and temperature (°C) (**b**) for the natural forest and rainfed crop system. Data are shown as the means ± SD (*n* = 3). Drought periods are noted by circles.

**Figure 2 microorganisms-11-02589-f002:**
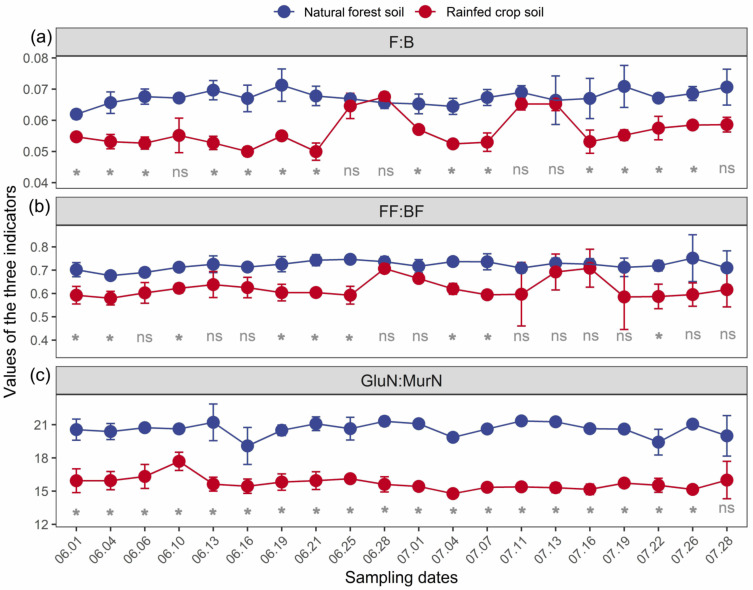
The F:B (**a**), FF:BF (**b**), and GlcN:MurN (**c**) dynamics of a natural forest system vs. rainfed crop system at 20 measuring time points. Data are shown as means ± SD (*n* = 3) and ns indicates not significant (*t*-test, *p* > 0.05). *, *p* < 0.05; F:B is the fungi to bacteria ratio; FF:BF is the fungivore to bacterivore ratio; GlcN:MurN is the glucosamine to muramic acid ratio.

**Figure 3 microorganisms-11-02589-f003:**
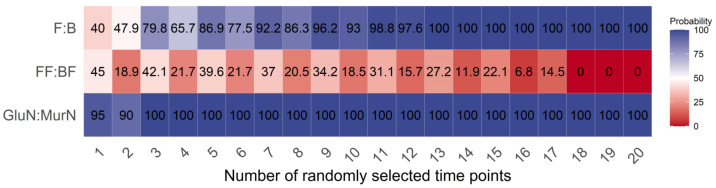
Effects of randomly selected measuring time point numbers on the reliability of each indicator.

**Figure 4 microorganisms-11-02589-f004:**
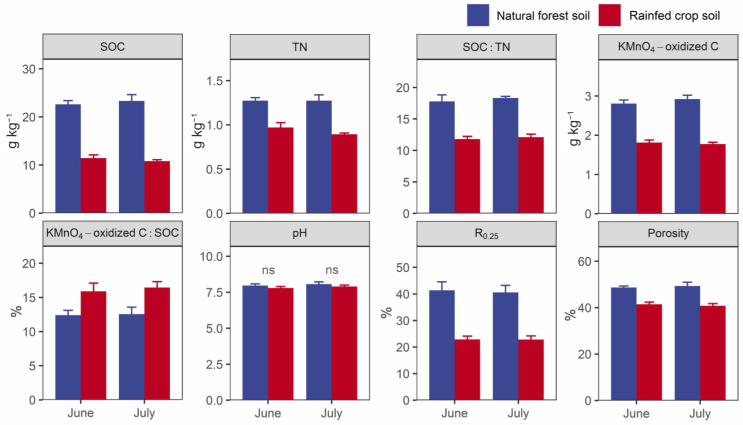
Selected key soil properties of the natural forest system vs. rainfed crop system in June and July. SOC: soil organic carbon; TN: total nitrogen; R_0.25_: proportion of >0.25 mm aggregate. Data are shown as means ± SD (*n* = 3) and ns indicates not significant (*t*-test, *p* > 0.05).

**Figure 5 microorganisms-11-02589-f005:**
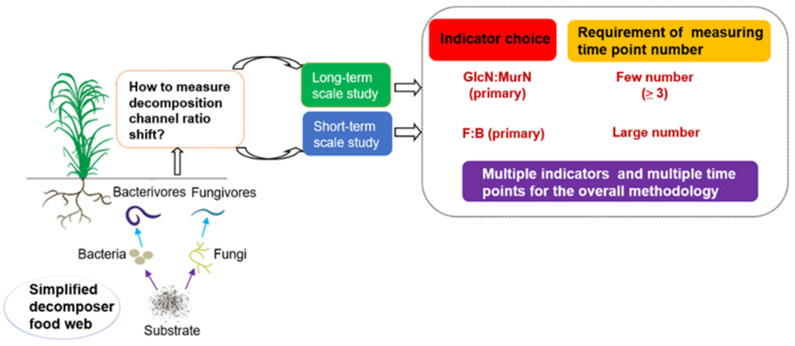
Conceptual suggestion for indicator selection and the requirement of measuring time point numbers for examining shifts in soil gross decomposition channel ratio. F:B is the fungi to bacteria ratio; GlcN:MurN is the glucosamine to muramic acid ratio.

## Data Availability

All data generated or analyzed during this study are included in this article.

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
