# Peer review of "Examining the Shift in the Decomposition Channel Structure of the Soil Decomposer Food Web: A Methods Comparison"

_microorganisms, 2023, doi:10.3390/microorganisms11102589_

Round 1

Reviewer 1 Report

The work of Wen Xing and co-authors is devoted to the development of a method to examine the shift of decomposition channel structure 2 of soil decomposer food web.

The authors carried out a series of measurements over two months and, based on the results obtained, showed which method was the most reliable. In general, the work corresponds to the profile of the journal Microorganisms.

I have no comments either in essence or regarding the English language.

the article may be accepted in its present form

Author Response

Point-by-Point Response

Reply to Reviewer #1

The work of Wen Xing and co-authors is devoted to the development of a method to examine the shift of decomposition channel structure of soil decomposer food web. The authors carried out a series of measurements over two months and, based on the results obtained, showed which method was the most reliable. In general, the work corresponds to the profile of the journal Microorganisms. I have no comments either in essence or regarding the English language. The article may be accepted in its present form.

Reply: Thanks a lot! We are very grateful to you for taking the time to review our manuscript and giving us positive comments.

Reviewer 2 Report

 I have read the paper carefully, it should be pointed out that the current research work benefits from an interesting topic and the resulting observations can be helpful. There are many major shortcoming that must be addressed before the review process can continue. So, this paper requires further refinement for publication.

The title doesn't seem appropriate.

The introduction needs a lot of work. There is no clear logical flow between sections.

Analytical methods are exposed in a cumbersome way. Please rewrite.

Line 82 Soil of the two sites is classified as a Haplic-Ustic Cambosol (FAO Classification). Cambosol?

Due to the poor quality of communication, the conclusions are not supported.

I am not an expert in the English language. There are several typographical and grammatical errors in the text, the manuscript should be revised in terms of grammatical rules by a native English speaker.

Author Response

Point-by-Point Response

Reply to Reviewer #2

1. I have read the paper carefully, it should be pointed out that the current research work benefits from an interesting topic and the resulting observations can be helpful. There are many major shortcomings that must be addressed before the review process can continue. So, this paper requires further refinement for publication. 

Reply: We very much appreciate the reviewer’s constructive comments and suggestions on our manuscript. We have revised the manuscript according to the suggestions from this reviewer. Please find the detailed response as below.

2. The title doesn't seem appropriate. 

Reply: We totally agree with the reviewer that the title is inappropriate. Following the reviewer’s suggestion, we have revised this title as “Examining the shift in decomposition channel structure of soil decomposer food web: Methods comparation” to be more suited to our research.

3. The introduction needs a lot of work. There is no clear logical flow between sections.

Reply: We fully agree with the reviewer that the logical flow between sections in introduction was not clear enough in the previous version of our manuscript. According to the reviewer’s concerns, we have revised and restructured all sections in the introduction.

In the first paragraph, compared to the previous manuscript, we first clearly introduced the concepts of the fungal- and bacterial-based energy channels in soil decomposer food web. We then stated the ecological significances of fungal- and bacterial-based energy channels for agricultural production practices, or environmental protection programs. Finally, we pointed out that despite many previous studies used the abundance or biomass of microbial and nematode functional groups and their metabolites to assess the structures of soil decomposer food webs, challenges remain for selecting the appropriate indicators to accurately examine the shifts in soil gross decomposition channels.

In the second paragraph, we introduced that three types of ecological indicators based on: i) consumers; ii) predators; and iii) metabolites are commonly used to reflect the shifts of soil decomposition channel structure. We then introduced the ecological mechanisms of the fungi to bacteria (F:B) ratio, the fungal- to bacterial-feeding nematode abundance (FF:BF) ratio, and the glucosamine to muramic acid (GlcN:MurN) ratio as the critical indicators to explain how changes in fungal- and bacterial-based energy channels in soil decomposer food web. Finally, we concluded that although these indicators have been extensively used either individually or in combination in ecological studies, their reliability and the contexts of their applications toward the interpretation of soil food web characteristic have not been fully assessed.

In the third paragraph, we introduced two uncertainties regarding the reliability and application of these indicators (F:B, FF:BF, and GlcN:MurN ratios) for interpreting soil food web structure. Firstly, there is no unified strategy for the selection of suitable indicators to infer the shifts in decomposition channel ratios of the soil food web. Compared to the previous manuscript, we analyzed in detail that the reliability of F:B, FF:BF, and GlcN:MurN ratios could vary in predicting the long- and short-term changes in soil decomposition channel structure. Secondly, the numbers of measuring time points for both biotic and abiotic indicators are often limited and random across different studies. We provided a detailed statements that the numbers of measuring time points for biotic indicators (F:B and FF:BF ratios) and abiotic indicator (GlcN:MurN ratio) in predicting the long- and short-term changes in soil decomposition pathways may differ. Consequently, we suggested that the indiscriminate use of these different methodological indicators may lead to the misinterpretation of anthropogenic impacts on soil biological processes, and further, incorrect management decisions.

In the fourth paragraph, we clearly stated the research questions and the hypotheses.

Please see our revised Introduction section. Hope these revisions can meet the requirements of reviewer.

4. Analytical methods are exposed in a cumbersome way. Please rewrite. 

Reply: Thanks for this valuable comment and suggestion! We agree with the reviewer that the analytical methods are exposed in a cumbersome way. And thus, we have simplified this section, and delete some redundant contents in our revised manuscript (line 191-196).

5. Line 82 Soil of the two sites is classified as a Haplic-Ustic Cambosol (FAO Classification). Cambosol? 

Reply: We apologize for this mistake. We have revised the “Cambosol” as “Cambisol”.

6. Due to the poor quality of communication, the conclusions are not supported. 

Reply: We very much appreciate the valuable comments from the reviewer. As the reviewer noted, the conclusions were not supported by the poor quality of communication in our previous manuscript. In this revised manuscript, we firstly summarized the comparations about the reliability of three indicators in assessing the shift of decompose channel structure, based on a natural forest and rainfed crop system with over 60 years cultivation. Secondly, we stated the basic limitations of our research. Finally, we provided the guidelines for future research (line 353-370).

7. I am not an expert in the English language. There are several typographical and grammatical errors in the text, the manuscript should be revised in terms of grammatical rules by a native English speaker.

Reply: Thanks for this important suggestion! Edited by a native English-speaking professional colleague, we have thoroughly checked and corrected the typographical and grammatical errors in this revised manuscript.

Reviewer 3 Report

The manuscript entitled " How to examine the shift of decomposition channel structure of soil decomposer food web?" is interesting and has the potential to interest readers. The issues presented in this manuscript are consistent with the topics of the Journal "Microorganisms". The results have been well described, and the conclusions correspond to the research goal set by the authors.

The manuscript requires some improvements before publication in the Journal " Microorganisms".

1.  The "Introduction" chapter requires some additions. Namely, the authors should write about the role of bacteria and fungi in shaping soil quality and why they are a good indicator of soil quality. Thus, the purpose of the research will be better justified. Moreover, they should clearly state research hypotheses, which will then be verified.

2. In the "Conclusions" chapter, try to write what are the basic limitations of your research and what are the guidelines for future research.

3.    Please, be sure that all the references cited in the manuscript are also included in the reference list and vice versa with matching spellings and dates. 

Author Response

Point-by-Point Response

Reply to Reviewer #3

1. The manuscript entitled " How to examine the shift of decomposition channel structure of soil decomposer food web?" is interesting and has the potential to interest readers. The issues presented in this manuscript are consistent with the topics of the Journal "Microorganisms". The results have been well described, and the conclusions correspond to the research goal set by the authors. The manuscript requires some improvements before publication in the Journal "Microorganisms".

Reply: We very much appreciate the positive comments and the helpful suggestions of the reviewer. We have revised the manuscript according to the suggestions from this reviewer. Please see our responses to the suggestions and comments below.

2. The "Introduction" chapter requires some additions. Namely, the authors should write about the role of bacteria and fungi in shaping soil quality and why they are a good indicator of soil quality. Thus, the purpose of the research will be better justified. Moreover, they should clearly state research hypotheses, which will then be verified.

Reply: Thanks for these valuable suggestions! It inspires us a lot! As the reviewer suggested, we have provided more details to emphasize the role of bacteria and fungi in shaping soil quality and why they are a good indicator of soil quality (line 31-49): “Two energy channels of the soil food web can explain how changes in decomposition processes of detritus impact ecosystem processes. First, the bacterial-based channel includes bacteria as the primary decomposer of substrate resources, which mainly transfers the flow of energy and material through bacterial-feeding fauna and their predators at high trophic levels. In general, the bacterial channel primarily occurs in nitrogen-rich soils that contain easily decomposable substrates, and promotes the nutrient mineralization rate and biological activities. Second, the fungal-based channel with fungi as the primary decomposer, which plays a key role in the exchange of matter and energy between substrate resources and the high trophic levels of fungi-feeding fauna and their predators. The fungal channel derives most of its energy from recalcitrant organic matter (e.g., high carbon-to-nitrogen ratio) or acidic soil and is linked to the highly conservative cycling of nutrients and increased soil carbon and nitrogen retention. The importance of the fungal-based channel in the soil food web often increases following land abandonment or vegetation restoration, while the bacterial-based channel becomes increasingly important following intense disturbances, nutrient enrichment, and intensive farming. Therefore, assessing the relative importance of fungal- vs. bacterial-based energy channels in soil decomposer food webs may be an explicit management objective for agricultural production practices, or environmental protection programs”.

We have also added the research hypotheses, which was verified in the Discussion section. Specifically, We test two hypotheses (line 108-117): “(H1) The GlcN:MurN ratio provides a time-integrated measure of the fungal to bacterial channel ratio to predict changes between the two ecosystems over the long-term, while the F:B and FF:BF ratios offer more specific data on the decomposition channel ratios to predict their changes over the short-term. (H2) A few measuring time points for the GlcN:MurN ratio are adequate for the detection of long-term shifts in the soil decomposition channel ratio between a natural forest and rainfed crop system, since amino sugars are negligibly decomposed in nature. In contrast, multiple time points are required for measuring soil decomposition channel ratios over short-term study periods, as many soil organisms are very sensitive to fluctuations in environmental conditions”.

3. In the "Conclusions" chapter, try to write what are the basic limitations of your research and what are the guidelines for future research. 

Reply: Thanks a lot for those valuable suggestions! As the reviewer suggested, we have provided more details to emphasize the basic limitations of our research and the guidelines for future research (line 359-371): 

“Due to the limitations of the applied indicators for this study, the reliability of other indicators estimated by the PLFA, growth rate, carbon metabolism, and specific nematode functional guilds require further assessment. We suggest the use of diverse indicators and adoption of multiple measuring time points for an overall methodology. First, future research should assess the mechanisms through which changes in decomposition channel structure relate to the question of interest. Second, the most relevant strategy for determining fungal vs. bacterial channel dominance should be selected. Specifically, to interpret changes in decomposition pathways over long-term scales, the amino sugar index is likely a more applicable indictor with few required measurement time points. However, to interpret changes in decomposition pathways over short-term scales, biotic indicators such as the fungi to bacteria ratio using multiple measurement time points, are likely a prudent choice to reflect changes in the soil decomposer food web”.

4. Please, be sure that all the references cited in the manuscript are also included in the reference list and vice versa with matching spellings and dates. 

Reply: Thanks for this important suggestion! We have carefully checked all the references cited in the manuscript and ensured them to be included in the reference list. We also checked whether the spellings and dates were matched.

Round 2

Reviewer 2 Report

The authors improved the quality of the paper by accepting most of the suggestions recommended by me. Now I can recommend the paper for publication as it is.